# Cord Blood Adductomics Reveals Oxidative Stress Exposure Pathways of Bronchopulmonary Dysplasia

**DOI:** 10.3390/antiox13040494

**Published:** 2024-04-20

**Authors:** Erika T. Lin, Yeunook Bae, Robert Birkett, Abhineet M. Sharma, Runze Zhang, Kathleen M. Fisch, William Funk, Karen K. Mestan

**Affiliations:** 1Department of Pediatrics, Division of Neonatology, University of California San Diego, 9500 Gilman Drive, La Jolla, CA 92093, USA; 2Department of Preventive Medicine, Northwestern University, 680 North Lake Shore Drive, Suite 1400, Chicago, IL 60611, USA; yeunook.bae@northwestern.edu (Y.B.);; 3Department of Pediatrics, Division of Neonatology, Northwestern University, Chicago, IL 60611, USA; 4Department of Obstetrics, Gynecology and Reproductive Sciences, University of California San Diego, La Jolla, CA 92093, USA; kfisch@health.ucsd.edu

**Keywords:** addition products, neonate, pre-eclampsia, chorioamnionitis, intrauterine growth restriction

## Abstract

Fetal and neonatal exposures to perinatal oxidative stress (OS) are key mediators of bronchopulmonary dysplasia (BPD). To characterize these exposures, adductomics is an exposure science approach that captures electrophilic addition products (adducts) in blood protein. Adducts are bound to the nucleophilic cysteine loci of human serum albumin (HSA), which has a prolonged half-life. We conducted targeted and untargeted adductomics to test the hypothesis that adducts of OS vary with BPD. We studied 205 preterm infants (≤28 weeks) and 51 full-term infants from an ongoing birth cohort. Infant plasma was collected at birth (cord blood), 1-week, 1-month, and 36-weeks postmenstrual age. HSA was isolated from plasma, trypsin digested, and analyzed using high-performance liquid chromatography–mass spectrometry to quantify previously annotated (known) and unknown adducts. We identified 105 adducts in cord and postnatal blood. A total of 51 known adducts (small thiols, direct oxidation products, and reactive aldehydes) were increased with BPD. Postnatally, serial concentrations of several known OS adducts correlated directly with supplemental oxygen exposure. The application of large-scale adductomics elucidated OS-mediated pathways of BPD. This is the first study to investigate the “neonatal–perinatal exposome” and to identify oxidative stress-related exposure biomarkers that may inform antioxidant strategies to protect the health of future generations of infants.

## 1. Introduction

Bronchopulmonary dysplasia (BPD) remains the most common chronic lung disease of infancy, affecting over half of extremely preterm (EPT) infants worldwide [1]. Severe BPD is characterized by chronic supplemental oxygen exposure, mechanical ventilation dependence, prolonged and recurrent hospitalizations within the first year of life, and long-term cardiopulmonary, metabolic, and neurodevelopmental problems [2,3,4,5,6]. Putative early-life exposures include pre-eclampsia and uteroplacental insufficiency during pregnancy, and hyperoxia in the early postnatal period [7]. Other maternal and neonatal exposures contributing to BPD have yet to be identified, but an important scientific premise is that these exposures collectively contribute to a common putative pathway of elevated oxidative stress (OS) [8].

Reactive oxygen and nitrogen species (ROS/RNS) are central to BPD pathophysiology as they directly damage DNA and the proteins necessary for lung growth and repair [9,10,11]. These electrophiles enter the blood from absorption in the lungs or via metabolism in the liver and other tissues, from oxidation of lipids and other molecules, and from inflammation associated with co-existing disease states. Once entering the blood, electrophiles react with available proteins to form addition products, or “adducts” [12]. When bound to stable proteins such as human serum albumin (HSA), these adducts become significantly more stable (28 days in circulation) than the scavenged reactive electrophiles [12], and thus may serve as more reliable biomarkers of exposure in chronic disease states, such as BPD.

The HSA-Cys^34^ residue (Figure 1) is a sentinel nucleophilic hotspot for protein adducts that accounts for 80% of the antioxidant capacity of serum/plasma [13]. HSA-Cys^34^ adducts potentially play a pivotal role in understanding human disease arising from exposures to electrophilic species [12,14]. Until recently, studies addressing HSA-Cys^34^ adducts have been mainly focused on untargeted assays. While required for discovering unknown HSA-Cys^34^ adducts, untargeted approaches typically have lower analytical sensitivity than targeted assays. Our recently developed targeted assays can simultaneously quantify large panels of HSA-Cys^34^ adducts with higher detection rates than untargeted assays [12,14,15]. The application of untargeted adductomics in an EPT infant population would provide new opportunities to investigate the “neonatal exposome” as it relates to BPD. Once identified, oxidative stress-related exposure pathways and novel biomarkers could be further characterized that predict disease, track meaningful responses to perinatal and NICU exposures, and inform the development of therapies to prevent BPD. 

We employed novel exposomic approaches in a single-center study of EPT and healthy full-term (FT) infants. Our primary objective was to identify cord blood HSA-Cys^34^ adducts that are associated with BPD. In postnatal blood from EPT infants, we sought to further identify HSA-Cys^34^ adducts that correlate with cumulative supplemental oxygen (CSO) exposure in the postnatal period. We hypothesized that the adductomics profiles would elucidate perinatal and postnatal mechanisms of OS-mediated BPD.

## 2. Materials and Methods

### 2.1. Study Design and Patient Sample

Patients were drawn from an ongoing birth cohort study conducted at Prentice Women’s Hospital (Chicago, IL, USA) from 2008 to 2021. As shown in Figure 2, there were two arms of the study: (1) The cord blood arm comprised 210 births (159 EPT and 51 FT) with corresponding archived cord blood plasma samples which were drawn from the larger birth cohort. A total of 79 BPD and 80 non-BPD infants were included in the final analysis, in addition to 51 healthy FT infants (born at ≥37 completed weeks without newborn complications) who were enrolled over the same time period. Informed consent was obtained from mothers prior to participation. The details of enrollment and sample/data collection, processing, and archive in the parent cohort have been published previously and are described elsewhere [16,17,18]. (2) The peripheral blood arm included 46 EPT infants who were followed prospectively from 2020 to 2021 with scavenged postnatal blood samples recovered from scheduled blood draws performed at 1-week, 1-month, and 36-weeks postmenstrual age (PMA). The samples were linked to clinical data throughout each patient’s NICU hospitalization, including respiratory data and cumulative supplemental oxygen exposure (see below). Informed consent was waived under a scavenged biospecimen protocol that was linked to coded data that were deidentified upon NICU discharge. There was no patient overlap between cord blood and peripheral blood arms.

### 2.2. Clinical Data Collection

Maternal and infant information was extracted from electronic medical records using standardized protocols and automated programs in consultation with Northwestern’s Enterprise Data Warehouse (EDW). This included detailed information on antepartum, intrapartum, and pregnancy outcomes, and serial data on infant hospitalization course (see below) for each mother–infant dyad enrolled in the birth cohort. Maternal pre-eclampsia and other hypertensive disorders of pregnancy (eclampsia, and hemolysis, elevated liver enzymes, and low platelets (HELLP) syndrome) were defined according to American College of Obstetrics and Gynecology (ACOG) criteria. Chorioamnionitis was defined as having either clinical evidence according to ACOG criteria or histologic evidence of acute inflammation by placental pathology exam [19,20,21,22]. Small for gestational age (SGA) was defined as birth weight < 10th percentile for GA based upon Fenton growth curves for premature infants [23].

Detailed NICU data were collected on all EPT infants. This included serial information related to the primary outcome of BPD, which was defined using the modified NIH criteria [24] and included Grade I, II, III, or III(A) as determined at 36 weeks PMA. Neonatal complications of necrotizing enterocolitis (NEC), intraventricular hemorrhage (IVH), retinopathy of prematurity (ROP), and BPD-associated pulmonary hypertension (BPD-PH) were defined using our standardized protocols for the cohort, as previously published [16,18,25]. Respiratory support modes and FiO_2_ values at 8 h intervals (07:00, 15:00, and 23:00) were captured using automated electronic medical record algorithms. The calculation of cumulative supplemental oxygen (CSO) was adapted from a published algorithm as the sum of average daily supplemental oxygen (FiO_2_–0.21) over the time interval leading up to blood collection at 1-week, 1-month, and 36-weeks PMA [26]. CSO values at each timepoint were divided into quartiles and categorized for each patient as low (<25th percentile of the distribution), mild (25–50th percentile), moderate (51–75th percentile), and high CSO (>75th percentile of the distribution).

### 2.3. Cord Blood and Postnatal Peripheral Blood Collection

Umbilical venous cord blood was collected at birth into ethylenediaminetetraacetic acid (EDTA) tubes, aliquoted, and separated for plasma by tabletop centrifugation at 3000 rpm at 4 °C for 10 min. The plasma was archived at −80 °C until assay. Left-over peripheral blood plasma samples were retrieved from the Northwestern Memorial Hospital chemistry lab within 24 h after routine patient blood draws at 1-week, 1-month, and 36-weeks PMA, and stored at −80 °C until assay. 

### 2.4. Plasma Adductomics

**Chemical Reagents.** EDTA (BioUltra, anhydrous, ≥99%), triethylammonium bicarbonate buffer (TEAB) (1 M, pH 8.5), sodium hydroxide (BioXtra, ≥98%), HSA (lyophilized powder, ≥96%, agarose gel electrophoresis), and trypsin (from porcine pancreas, lyophilized powder, 1000–2000 BAEE units/mg solid) were obtained from Sigma-Aldrich (St. Louis, MO, USA). Formic acid (Optima LC/MS grade), acetonitrile (Optima LC/MS grade), and methanol (HPLC grade grade) were purchased from Fisher Scientific (Fair Lawn, NJ, USA). T3 peptide with the sequence ALVLIAFAQYLQQCPFEDHVK (>95%, LCMS Pure, 2433 g/mol) was custom synthesized by the Peptide Core in Northwestern University (Chicago, IL, USA). Here, it is important to note that the T3 peptide contains Cys^34^, which can promote the formation of protein adducts. All of the stock solutions and samples were prepared using deionized (DI) water (resistivity > 18.2 mΩ-cm) prepared with a PureLab purification system (ELGA LabWater, Woodridge, IL, USA).

**Isolation of HSA from Plasma**. A total of 60 µL of a mixture containing 50% methanol and 50% DI water was added to 5 μL aliquots of plasma to precipitate unwanted proteins. The mixture was mechanically shaken for 30 min followed by centrifugation for 15 min (14,000 rpm, 4 °C). Thirty microliters of the resulting supernatant was then transferred to protein digestion tubes (MT-96, Pressure Biosciences Inc., South Easton, MA, USA) and diluted with 120 μL of digestion buffer (50 mM triethylammonium bicarbonate and 1 mM ethylenediaminetetraacetic, pH 8.0). A total of 3 μL of 10 μg/μL trypsin (1:30 *m*/*m* trypsin/HSA) was added and protein digestion was performed using a Pressure Biosciences™ Barozyme HT48 system. After digestion, 3 μL of 10% (*v*/*v*) formic acid in DI water was added to each sample and briefly centrifuged at 14,000 rpm. A 40 μL aliquot was then mixed with 80 μL of 2% acetonitrile and 0.2% formic acid (*v*/*v*) in DI water and centrifuged at 14,000 rpm for 15 min. One-hundred microliter aliquots of supernatant were then transferred to borosilicate silanized glass vials (Microsolv Technology Corporation, Leland, NC, USA) prior to analysis. 

**Untargeted Adductomics.** Untargeted adductomics were initially performed on 16 cord blood plasma samples across a gestational age spectrum from 32 to 40 weeks. This served to identify all known adducts with corresponding structures as well as to detect unknown adducts with new organic structures for analysis. Once all adducts were validated, targeted adductomics results could be compared to validated known and unknown adducts from untargeted adductomics.

The discovery experiments were performed using an Orbitrap LTQ Velos Elite Orbitrap (Thermo Fisher, San Jose, CA, USA) equipped with a Dionex Ultimate 3000 RSLC nano system. A total of 5 µL of HSA protein digest was loaded onto a trap column (5 mm × 0.2 mm i.d., Thermo PepSwift™ Monolithic capillary column) with a flow rate of 5 µL/min for 5 min. A Thermo PepSwift™ Monolithic capillary analytical column (250 mm × 0.1 mm i.d.) with a custom nanospray source and a stainless-steel emitter was used for peptide separation (outer diameter 150 um; inter diameter 30 um). Solvent A contained 0.1% formic acid in LC/MS grade water and solvent B contained 0.1% formic acid in LC/MS grade acetonitrile. A 750 nl/min flow rate was used with the following gradient conditions: 0–7 min at 2% solvent B; 7–40 min, linear gradient from 2–40% B; 40–45 min, 98% B; 45–50 min, 2% B, with a total run time of 50 min. High-resolution data files were manually inspected using the Xcalibur™ Software (Version 4.1.50) to identify new adduct features. Total ion chromatograms (TICs) were examined stepwise in 0.088 min increments across retention times ranging from 20 to 60 min. Three *b* ions—*b*_3_^+^ (*m*/*z* = 284.42), *b*_4_^+^ (*m*/*z* = 397.63), and *b*_5_^+^ (*m*/*z* = 510.84)—were used to identify T3 peptides containing Cys^34^ adducts. A second series of three *y*^+^-series ions, including *y*_15_^+^ (*m*/*z* = 973.92 + α), *y*_16_^+^ (*m*/*z* = 1009.47 + α), and y_17_^+^ (*m*/*z* = 1066.07 + α), were then used to determine the added mass corresponding to new adduct features. Newly identified adduct features were then added to our targeted adductomics panel in subsequent experiments for adduct quantitation. 

**Targeted Adductomics.** Targeted adductomics was performed using an Agilent 6490 QqQ with an iFunnel electrospray source coupled to an Agilent 1260 Infinity HPLC system (Agilent Technologies, Santa Clara, CA, USA). An Agilent Poroshell 120 EC-C^18^ LC column was used for peptide separation (50 × 3 mm, 2.7 μm). Ten microliters of HSA digest was injected per sample and multiple reaction monitoring (MRM) was used to monitor precursor ions corresponding to triply charged T3 adducted peptides (y_15_^2+^, y_16_^2+^, and y_17_^2+^). Solvent A consisted of 0.1% formic acid in LC/MS grade water, and solvent B consisted of LC/MS grade acetonitrile. Adduct quantification and data processing are described in detail in previously published studies [27,28,29]. Raw data files were imported to Skyline (Ver. 21.1) with a complete set of transitions for each run [15]. Peaks were manually selected using 3 transitions based on nominal mass and retention time (y_15_^2+^, y_16_^2+^, and y_17_^2+^). Quantitation was based on the summed peak areas of the three transitions, normalized to a housekeeping peptide (HKP) [14]. The HKP is a doubly charged precursor ion adjacent in sequence to the T3 tryptic peptide (^41^LVNEVTEFAK^50^), which is used to normalize adducts concentrations to the amount of HSA in each sample [15]. All statistical analyses were performed using adduct peak areas/HKP peak area (PAR). PAR values were also converted to pmol of adduct/mg of HSA using two external calibration curves to estimate adduct concentrations (see Appendix A).

### 2.5. Statistical Analysis

Continuous data were analyzed using parametric or non-parametric tests, where appropriate. Categorical data were compared using Fisher’s exact or Chi-square tests. All adduct concentrations were non-normally distributed. To take this into account, adduct concentrations were scaled by log-transformation and normalized with Trimmed Means of M-Values (TMM) normalization to account for concentration variations between samples (TMM normalization excludes a percentage of upper and lower values, takes a trimmed mean, and generates scale factors to differentiate between samples). Mean adduct concentrations were analyzed with R software (Ver. 4.0.3) using the Bioconductor packages edgeR and limma [30]. Linear regression models were used to identify differentially expressed adducts by relevant clinical metadata variables between BPD, non-BPD, and healthy full-term (FT) infant controls. Statistical significance was based on a Bonferroni-adjusted *p*-value < 0.05 *P_adj_*) and significance thresholds for adductomics results are reported using *P_adj_* < 0.05 and *P_adj_* < 0.001.

## 3. Results

### 3.1. Characteristics of the Patient Cohort

Figure 2 illustrates the distribution of all 256 infants included in this study. Table 1 shows the baseline demographics and clinical characteristics of the patient sample. Of the 205 EPT infants, 102 infants had BPD and 103 did not, according to the study definition for BPD [24]. The three groups (FT vs. non-BPD vs. BPD) differed by maternal age and antenatal diagnosis of intrauterine growth restriction (IUGR), with a higher rate of IUGR in the BPD group (*p* < 0.001). Compared to non-BPD preterm infants, BPD infants had a lower gestational age (GA), a lower birth weight (BW), and lower APGAR scores, and were more likely to be SGA. Co-morbidities of NEC, IVH, ROP, and BPD-PH were increased with BPD.

### 3.2. Cord Blood Adductomics

Among the 210 infants, a total of 105 adducts were detected. Of these, 49 were unknown adducts and 56 were previously annotated (known) adducts. All adducts identified (assigned A001 to A105 for this study) and putative nomenclature and characteristics are shown in Appendix A. Samples were analyzed in duplicate and averaged across runs. The median of normalized concentrations for each adduct according to the study groups are shown in Appendix A. Previously annotated adducts were also assigned to functional categories based upon their chemical class [14,31,32,33,34,35,36], with relative abundance shown in Figure 3. 

#### 3.2.1. Known OS Adducts in Cord Blood Were Associated with Extreme Prematurity

We compared the 56 previously annotated (known) adduct concentrations according to EPT versus FT status (Figure 4A). There were 50 known adducts that were increased with EPT (Figure 4B). In addition, unmodified T3 (A006) was elevated with EPT, with 1.2-fold increase when compared to the FT group (*P_adj_* < 0.001).

#### 3.2.2. Known OS Adducts Were Associated with Peripartum Exposures of Prematurity

To further explore the role of in utero exposures that might contribute to oxidative stress, we conducted subsequent analyses of the EPT group with respect to major complications preceding preterm birth. We focused specifically on clinical evidence of maternal pre-eclampsia and chorioamnionitis. Of the 159 EPT infants, 24 had exposure to maternal pre-eclampsia. Any preterm infant with chorioamnionitis was excluded and we confirmed that there were no infants with both pre-eclampsia and chorioamnionitis. With pre-eclampsia alone, we found a 1.5-fold increase in Cys^34^ sulfinic acid (A012; *P_adj_* = 0.001). In addition, two adducts had decreased levels with pre-eclampsia (at least 1.1 FC; A005 and A006; *P_adj_* < 0.05). In EPT infants exposed to maternal chorioamnionitis (N = 19), we found decreased levels of multiple small thiols (S-Addition of CysGly (A051), S-Cys (A040), S-Addition of hCys (A044), S-Additions of Cys (NH2→OH; A041), Na adduct of S-Cys (A047), S-Addition of Cys, methylation (A045), and S-Sodiation (A010; at least 1.1 FC; *P_adj_* < 0.05).

#### 3.2.3. Known OS Adducts Were Associated with BPD and BPD-Associated Pulmonary Hypertension

A heatmap of relative concentrations among the 159 EPT infants, according to major categories of known adducts, is shown in Figure 5A. The same known adducts found in the EPT versus FT analysis were associated with BPD versus non-BPD (≥1.1 FC; *P_adj_* < 0.05), in addition to Cys^34^→Oxoalanine (A004), with a 1.4-fold increase with BPD (*P_adj_* < 0.001) (Figure 5B). We also performed a subgroup analysis comparing levels from BPD infants with PH at 36 weeks (BPD-PH; N = 25) versus BPD without PH (BPD only; N = 54). Fifty-four adducts were decreased with BPD-PH versus BPD only (*P_adj_* < 0.05) (Figure 5C). The adduct most notably decreased with BPD-PH was a dehydrated form of Cys^34^ sulfinic acid plus methylation (not on Cys^34^) (A012), with a 1.7-fold decrease (*P_adj_* < 0.001). As shown in Figure 5A, infants with late PH (≥36 weeks PMA) with or without BPD tended to have lower adduct concentrations, except for one patient with BPD-PH (born at 27 weeks) who was exposed to chorioamnionitis in utero and died at 36-weeks PMA.

### 3.3. Peripheral Blood (Postnatal) Adductomics

Characteristics of the 46 EPT infants in the postnatal arm were similar to the cord blood arm (Figure 2). In total, 39 infants had peripheral blood samples at 1 week, 40 infants at 1 month, and 28 infants at 36 weeks. The median age at blood collection for the three timepoints was similar between groups (Figure 2). Reasons for missed samples included incomplete recovery of scavenged blood, either because no sample was drawn at that timepoint for clinical indications or there was no remaining sample available for the study. Regardless, the rates of BPD did not differ significantly among infants with all three versus those with less than three consecutive samples. As listed in Appendix A, postnatal adductomics profiles were similar to cord blood in composition and range of concentrations. All 105 adducts that were identified in cord blood were detected in peripheral blood, with 49 unknown and 56 previously annotated adducts.

Figure 6 shows heatmaps of relative adduct concentrations at the three serial timepoints. Leading up to 1 week, non-significant trends of increasing adduct levels with increasing CSO appeared independent of BPD status. At 1 month, 51 known adducts were associated with high versus low CSO (fourth quartile versus first quartile; at least 2.1 FC; *P_adj_* < 0.05) (Figure 6B). Cys^34^→Oxoalanine (A004)—the adduct identified in cord blood with the highest fold-increase with BPD—was increased three-fold with high CSO exposure at 1 month (*P_adj_* < 0.05). At 36 weeks, nine known adducts were significantly decreased with high CSO (Figure 6C). Of these, Cys^34^ sulfinic acid (A012)—the adduct decreased in cord blood with BPD-PH—was decreased with high CSO at 36 weeks (*P_adj_* < 0.05).

In contrast to cord blood, there were no previously annotated adducts associated with BPD at any of the three timepoints. However, several unknown adducts correlated with CSO and BPD. Heatmaps of the relative concentrations of unknown adducts (Figure 7A) illustrate the high variability among individual patients across time with CSO exposure and BPD status. At 1 week, 48 unknown adducts were decreased with high CSO (fourth quartile; at least 2.5 FC; *P_adj_* < 0.05). At 1 month, 24 unknown adducts were increased with high CSO (≥4.3 FC; *P_adj_* < 0.001). Lastly, 45 unknown adducts directly correlated with BPD at 1 month (≥1.1 FC; *P_adj_* < 0.05; Figure 7B) but none were correlated at 1-week or 36-weeks PMA.

## 4. Discussion

We employed novel adductomics approaches to identify protein adducts in cord blood and peripheral blood associated with extreme prematurity and BPD. The adducts included in this analysis are implicated in intrinsic and extrinsic pathways of OS, based upon their distinct biochemical structures. In total, 105 adducts were identified, with certain adducts varying in concentration that correlated with the degree of prematurity, the development of BPD, and exposure to supplemental oxygen. These included thiols, direct oxidation products, and others which have been associated with environmental exposures and disease states such as tobacco smoke, air pollution, and certain cancers [27,32,37,38]. While the literature on the role of adductomics in understanding the human exposome is rapidly expanding, this is the first study to investigate the “neonatal–perinatal exposome” of extremely preterm infants to identify exposure biomarkers of BPD.

Notable findings from this study include the following: (1) unmodified T3 was increased in cord blood with EPT; (2) cord blood Cys^3^→Oxoalanine was increased with BPD; (3) Cys^34^ sulfinic acid was decreased with BPD-PH, but increased with pre-eclampsia; (4) in postnatal blood, several known adducts correlated with CSO, and several unknown adducts were increased with BPD. Heatmaps of relative concentrations suggest a trend toward lower levels of OS-related adducts in patients with pre-eclampsia and pulmonary hypertension, and increased levels with chorioamnionitis. 

Elevated OS is an important mediator in the pathogenesis of BPD. Relative hyperoxia at birth and in the neonatal period, due to supraphysiologic oxygen exposure, generates ROS/RNS that damage the immature developing lung, inhibit growth and repair, and lead to chronic inflammation and fibrosis [39,40]. The reduced antioxidant capacity inherent in premature infants contributes to BPD pathogenesis [8]. Despite the proposed use of antioxidants in human infants [41], as well as widespread strategies to reduce supplemental oxygen exposure in the postnatal period [42,43], the incidence of BPD remains unacceptably high and variable among infants born at ≤28 completed weeks [41]. The association between BPD and early pregnancy risk factors such as IUGR now suggests that peripartum exposures, in particular those resulting in chronic fetal hypoxia of placental dysfunction and pre-eclampsia, play important roles and may be independent of postnatal factors such as hyperoxia [44,45]. These peripartum processes are poorly understood, largely because there are few stable biomarkers of OS at birth to elucidate these mechanisms. 

Well-characterized OS biomarkers include byproducts of lipid peroxidation (8-isoprostane), oxidative DNA damage (8-hydroxy-2-deoxyguanosine), and others that capture the immediate biological state of ROS at the time of collection [46,47]. Unique to HSA is its nucleophilic hotspot to which a host of OS-related adducts binds. Unlike lipid peroxidation and DNA byproducts, HSA has a much longer half-life, with a mean residence time of about 1 month. Thus, HSA-Cys^34^ protein adductomics profiles from cord blood could serve as a molecular fingerprint of perinatal exposures in the weeks leading up to preterm delivery. These profiles captured at birth in cord blood, coupled with serial postnatal adductomics profiles in the weeks leading up to BPD diagnosis at 36-weeks PMA, provide the first report applying unbiased adductomics to a large cohort of NICU patients.

The patient sample represented a broad range of perinatal and postnatal characteristics, and included a sizable cohort of FT infants from a single center through which we could contrast the influence of extreme prematurity on adductomics profiles. Consistent with other preterm cohorts and the body of literature on BPD risk factors, lower GA and BW as well as SGA were associated with BPD (Table 1). Antenatal IUGR was a strong predictor of BPD, with 15.7% of BPD versus 1.9% of non-BPD infants having prenatally detected fetal growth restriction. 

Two key adducts associated with BPD were Cys^34^→Oxoalanine (A004) and Cys^34^ sulfinic acid (A012). A004, a protein truncation product that has been cited for its association with OS, was increased with BPD. In contrast, A012, a direct oxidation product, was significantly decreased with BPD-PH. In fact, the majority of known adducts increased with any BPD were decreased with BPD-PH. This suggests a dichotomy of OS pathways between BPD only and BPD-PH subphenotypes, in which BPD-PH may represent a more severe form of OS-mediated lung injury associated with a heightened antioxidant response in certain individuals, leading to a decrease in direct oxidation products bound to HSA-Cys^34^. In contrast to BPD-PH, A012 was increased with pre-eclampsia—a specific peripartum risk factor for BPD-PH [25]. This “flip” from increase to decrease suggests that a transition in antioxidant capacity and the presence of direct oxidation products occurs, perhaps with heightened ROS/RNS generated by the placenta in the peripartum period [48] and subsequent upregulation of antioxidant activity in the infant during the postnatal period leading up to BPD at 36 weeks [49]. These adaptive mechanisms may be programmed during fetal development and necessary for survival, as suggested in the outlier case of an infant with BPD-PH with very high concentrations of OS-related adducts at birth who was exposed to intrauterine infection (chorioamnionitis) and eventually died at 36 weeks PMA (Figure 5A). 

While the number of samples available for postnatal adductomics were relatively small, some trends were notable and consistent with the above speculations. At 1-month, but not at 1-week or 36-weeks PMA, a large proportion of known adducts were increased in infants with high CSO exposure in the weeks leading up to 1 month of age (Figure 6A). Again, there was a “flip” in the pattern in which only nine adducts varied according to CSO, and all nine were decreased in infants with high CSO in the weeks leading up to 36-weeks PMA (Figure 6B). Of note, A012—the adduct which decreased with BPD-PH—was among those identified with high CSO. 

Among the unknown adducts, a large majority were associated with high CSO and followed the trends of the known adducts in that they were largely decreased at 1 week, but increased at 1 month (Figure 7B). We speculate that a key transition in antioxidant activity may be occurring in the weeks preceding 1-month after birth that is mediated by CSO exposure, as 45 unknown adducts were increased at 1 month in BPD infants, but not at 1 week or 36 weeks. 

There are currently over 50 previously annotated adducts described in adults [14,15,27,32,33,34,35,36,37,38] and over 40 unknown adducts recently identified [14,31,32,33,34,35,36,38,50]. The adducts identified in our study have been associated with pathological disease states: Cys^34^→Oxoalanine is increased in individuals exposed to tobacco smoke, solid fuel, and benzene associated with colorectal cancer [37,38,51]. The unanticipated trend of decreased adduct levels in severe diseases such as BPD-PH has been observed in comparable human disease states. For example, in smokers, lower levels of direct oxidation products have been reported in association with smoking-induced hypoxia [14]. Cys34-sulfinic acid is decreased with air pollution exposure in chronic obstructive pulmonary disease (COPD) and ischemic heart disease (IHD) [33]. Liu and colleagues postulated that the increased OS of COPD and IHD generates greater production of Cys^34^ oxidation products. In turn, the hypoxia induced by severe lung disease, exacerbated by air pollution exposure, may dysregulate critical redox control pathways. Similar complex pathways may exist with perinatal chronic fetal hypoxia, postnatal hyperoxia, and severe BPD. For example, in cases of chronic fetal hypoxia associated with IUGR and placental insufficiency, the transition to relative hyperoxia at birth and in the postnatal period can trigger an OS cascade that overwhelms the compromised antioxidant system of the EPT infant and adapts to compensate for survival over time [8].

The majority of OS adducts were in small thiols, direct oxidation products, and reactive aldehydes groups (Figure 3). Small thiols, known for their antioxidant capacity, may be a response over time to high OS exposure. For instance, at 1 month of life, chronic high saturation of ROS from physiologic hyperoxia may activate antioxidant pathways. This is supported by the increase in *Cys^34^*→*Oxalanine*, a protein truncation product resulting from OS, with high CSO at 1 month of life. Finally, reactive aldehydes are associated with lipid peroxidation, a process that triggers cellular dysfunction or intrinsic pathways, like apoptosis, in the setting of elevated OS. Ogihara and colleagues reported that reactive aldehydes are elevated at birth in premature infants who develop BPD. These aldehydes may induce the overexpression of cytokines involved in lung fibrosis [52].

The limitations of this study included the relatively small patient sample for the large number of adducts analyzed, and the single-center cohort design—which provided uniformity of the NICU environment, but future multi-center studies are needed to assess the generalizability of our findings. Nevertheless, this is the largest study of its kind, involving over 200 EPT infants. It was necessary to combine a pre-existing cohort with banked cord blood samples with a prospective cohort to follow serial timepoints and CSO exposures. There were some patients for which we could not recover data for all three timepoints. However, the same sets of adducts were identifiable and measurable in scavenged peripheral blood as in prospectively collected cord blood during the previous 13 years. This supports the stability of plasma adducts in archived samples and the future utility of HSA-Cys^34^ adducts as biomarkers for EPT infants as very minimal blood volume is needed (5 µL of sample for a single run). Recent studies of adductomics performed on neonatal dried blood spots [27,29] provide promise for the development of more practical approaches to tracking these exposure biomarkers over time in critically ill neonates. Utilizing dried blood spots has the advantage of being minimally invasive and more widely available as a source of peripheral blood measurement in neonates, with comparable stability in recent clinical studies [53,54,55,56,57]. Given that plasma and DBS samples are processed very differently, further validation is needed in order to integrate adduct levels from different blood sources, timepoints, and collection methods.

Lastly, relative to other omics platforms in which biological pathway analyses are relatively more developed, the field of adductomics is still in its infancy. However, the findings of this study contribute to this rapidly growing field of exposure science. Validation studies employing synthetic chemistry work to characterize unknown adducts associated with BPD are important next steps. The investigation of known adducts in validation studies and experimental models is also warranted. Specifically, our understanding of the mechanistic links between developmental lung disorders such as BPD and specific maternal environmental exposures (e.g., toxins and pollutants), as well as other NICU exposures (e.g., diet and medications), could be enhanced by integrating adductomics into larger epidemiologic studies and future clinical trials. Additionally, HSA-Cys^34^ biomarkers can be utilized in diagnostics laboratories. For instances, Cys^34^ sulfinic acid (A012) can indicate the status of reactive oxygen species [27]. By training medical staff on how to pretreat blood samples (e.g., plasma or DBS), certain adducts, including Cys^34^ sulfinic acid (A012), can be used to measure oxidative-stress status.

## 5. Conclusions

In conclusion, the application of adductomics to elucidate exposure pathways of preterm birth is a novel and promising next step in BPD research. These exposure pathways could inform the development of antioxidant strategies to reduce the incidence and severity of BPD. Linked to other high-resolution and high-throughput technologies, investigating multiple omics could yield unprecedented findings and lead to innovative approaches in the identification, classification, and management of BPD and its subphenotypes. This study highlights the central and complex role of OS signaling and regulation in the pathogenesis of multi-factorial BPD.

## Figures and Tables

**Figure 1 antioxidants-13-00494-f001:**
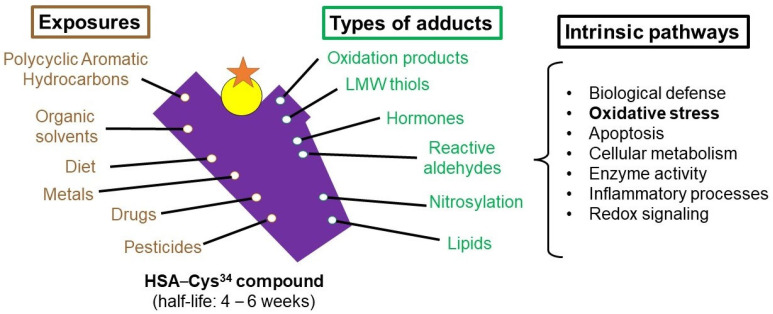
The human serum albumin–cysteine^34^ compound captures addition products (adducts) from variable exposures affecting downstream biological pathways. The purple shape represents the HSA protein. The yellow circle represents the Cys^34^ residue. The red star represents an adduct within the HSA-Cys^34^ compound. The green and brown terms indicate environmental exposures, which are mostly reactive and potentially toxic electrophiles. The black terms indicate potential adverse biological mechanisms, which can be induced by toxic electrophiles.

**Figure 2 antioxidants-13-00494-f002:**
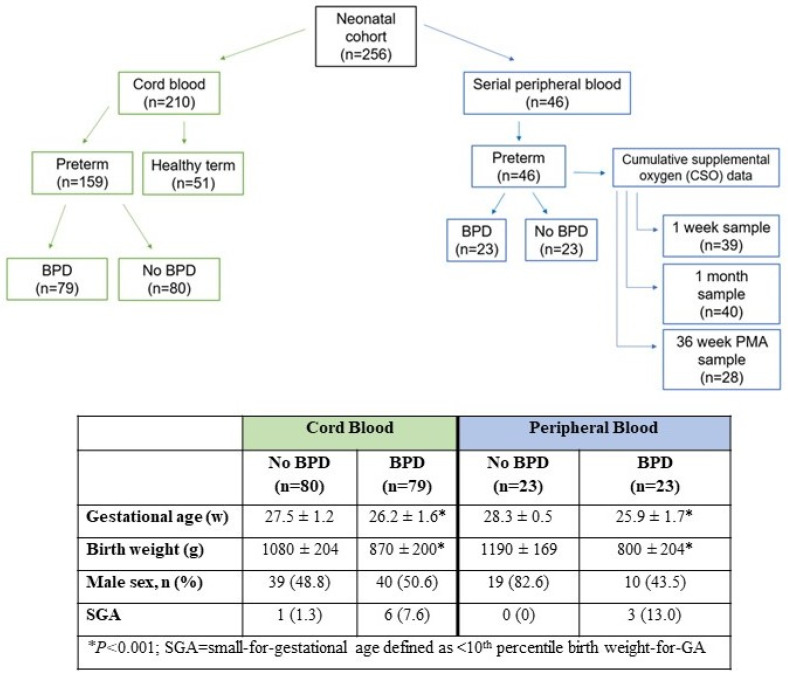
**A breakdown of the infant cohort by cord blood and peripheral blood groups.** Each serial peripheral blood sample was linked to respiratory data that included a calculation of cumulative supplemental oxygen (CSO). None of the infants from the cord blood arm (left) overlapped with the serial peripheral blood arm (right). As shown in the table, differences in baseline characteristics were similar between the two arms.

**Figure 3 antioxidants-13-00494-f003:**
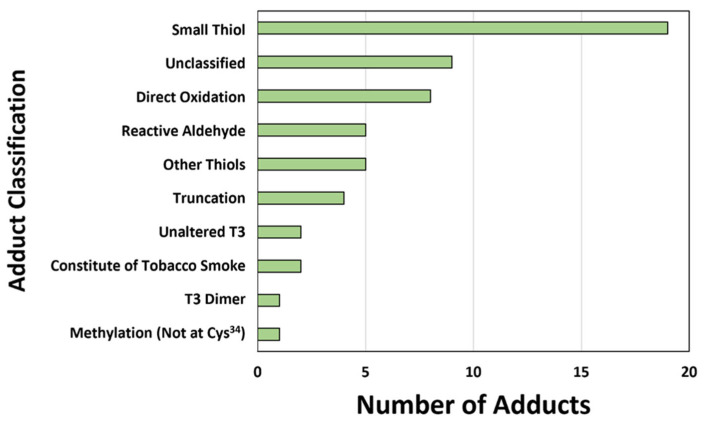
**Previously annotated adduct classifications.** Adducts were classified based on exogenous sources (e.g., tobacco smoke) and underlying endogenous processes (e.g., oxidative stress). Small thiols are Cys^34^ disulfides formed with small thiol molecules related to antioxidant capacity. Direct oxidation products are the addition of 1–3 oxygen molecules formed by reactions with Cys^34^ and ROS. Reactive aldehydes are reaction products formed through lipid peroxidation. Truncation products are reactions that result in the loss of mass at Cys^34^. Methylation (not at Cys^34^) refers to methyl groups added to the T3 peptide on amino acids other than Cys^34^. Unaltered T3 are T3 peptides in their reduced form, with no adducts on Cys^34^. Other disulfides are small thiols related to the microbiome and other sources not involved in antioxidant capacity. The constituents of tobacco smoke are exogenous adducts associated with tobacco smoke exposure. T3 dimers are disulfides formed between two T3 peptides.

**Figure 4 antioxidants-13-00494-f004:**
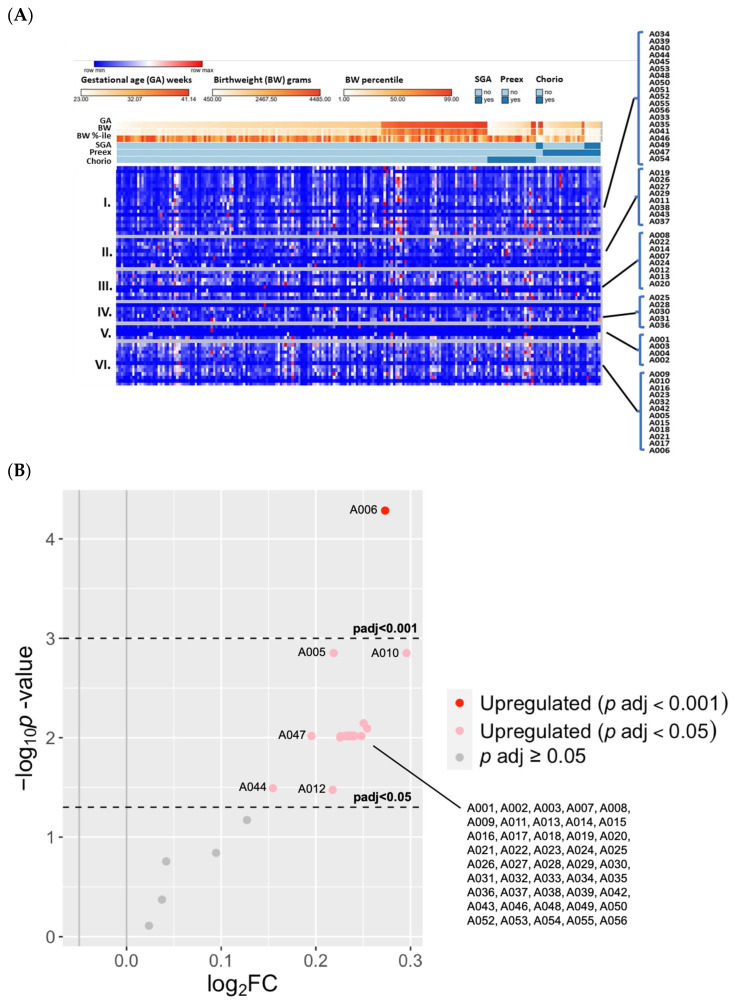
The differences in previously annotated (known) adduct concentrations according to gestational age and covariates of preterm birth. (**A**) A heatmap of relative concentrations of the 56 known adducts (listed on right), and grouped by (I) small thiols, (II) unclassified, (III) direct oxidation, (IV) reactive aldehydes, (V) truncation, and (VI) others. Heatmaps were generated using Morpheus https://software.broadinstitute.org/morpheus (accessed on 13 September 2023). (**B**) A volcano plot of upregulated known adducts with EPT versus FT infants. Significantly different adduct levels were filtered by a log fold-change of 2 and adjusted *p* values of *P_ad_*_j_ < 0.05 and *P_adj_* < 0.001.

**Figure 5 antioxidants-13-00494-f005:**
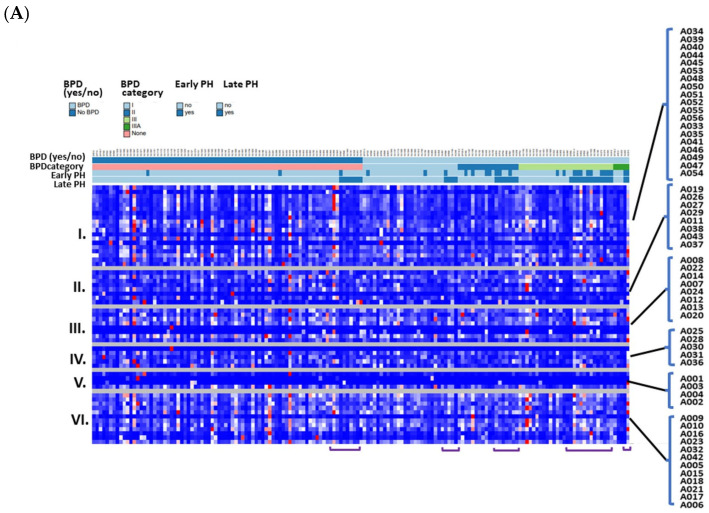
**The differences in known adduct concentrations according to BPD and pulmonary hypertension.** (**A**) A heatmap showing the relative concentrations according to BPD status and modified NIH categories (I-IIIA), and the presence of PH at ≤1-month (early PH) and ≥36-weeks PMA (late PH). Overall, the BPD group (right) appeared to have lower concentrations across all adduct classes. Infants with late PH, with or without BPD, tended to have lower adduct concentrations (purple brackets). One exception was a BPD-PH infant with relatively high adduct levels (far right) who presented with chorioamnionitis and died of bacterial sepsis. (**B**) A volcano plot comparing BPD and non-BPD controls. Significantly different adduct levels were filtered by a log fold-change of 2 and adjusted *p* values (*P_ad_*_j_ < 0.05 and *P_adj_* < 0.001). Most notable was Cys^34^→Oxoalanine (A004), a truncation adduct increased with BPD (*P_adj_* < 0.001). (**C**) A volcano plot comparing the known adducts according to BPD-PH versus BPD without PH (BPD only) infants. Most notable was a dehydrated form of Cys^34^ sulfinic acid plus methylation (A012), a direct oxidation product that decreased with BPD-PH (*P_adj_* < 0.001).

**Figure 6 antioxidants-13-00494-f006:**
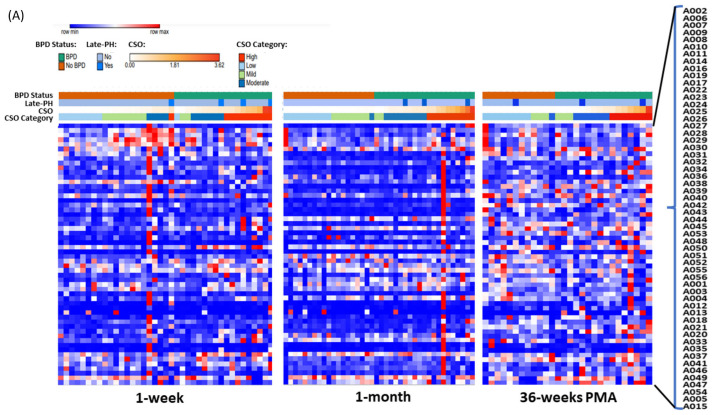
Known adducts detected in peripheral blood with cumulative supplemental oxygen (CSO) exposure and BPD. (**A**) A heatmap illustrating the variation in previously annotated adduct concentrations according to BPD and no BPD, and levels of CSO preceding 1-week, 1-month, and 36-weeks PMA. (**B**) At 1 month, there were 51 known adducts (red) associated with high CSO compared to low CSO (4th quartile versus 1st quartile, respectively; at least 2.1 FC; *P_adj_* < 0.05). (**C**) At 36 weeks, 9 adducts were significantly decreased (blue) with high CSO exposure. Grey dots indicate adducts that were not significant (*p*
> 0.05).

**Figure 7 antioxidants-13-00494-f007:**
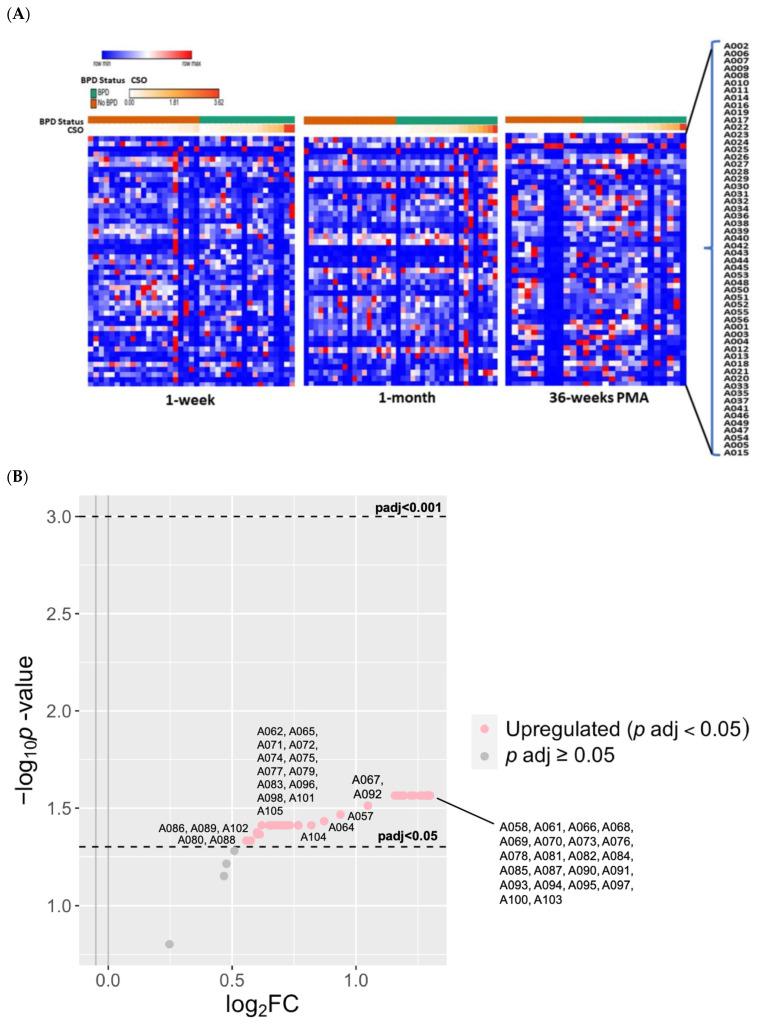
**Unknown adducts in peripheral blood with CSO and BPD**. (**A**) A heatmap showing the relative concentrations of unknown adducts according to BPD and CSO levels. In contrast to known adducts of oxidative stress, there was high variability among individual patients across time with CSO exposure and BPD status. (**B**) The volcano plot of the 49 unknown adducts at 1 month illustrates that 45 were significantly increased with BPD. Unknown 202.02 Da (A084) and Unknown 183.02 Da (A081) had a 2.5-fold increase with BPD (*P_adj_* < 0.05).

**Table 1 antioxidants-13-00494-t001:** Baseline demographics and clinical characteristics of the patient sample.

	Full Term (n = 51)	No BPD (n = 103)	BPD (n = 102)	*p* *
MATERNAL				
Maternal age, years, (mean ± SD)	34.6 ± 4.1	30.5 ± 5.6	32.3 ± 5.6	0.02
Maternal race, n (%)				
White	40 (78.4)	41 (39.8)	42 (41.2)	0.1
Black/African American	3 (5.9)	30 (29.1)	24 (23.5)	
Asian	1 (2.0)	5 (4.9)	7 (6.9)	
American Indian/Alaska Native	0 (0)	2 (1.9)	2 (2.0)	
Other	5 (9.8)	19 (18.4)	16 (15.7)	
Declined/Unknown	2 (3.9)	6 (5.8)	11 (10.8)	
Ethnicity, n (%)				
Hispanic or Latino	10 (19.6)	25 (24.3)	18 (17.6)	0.47
Not Hispanic or Latino	41 (80.4)	73 (70.9)	75 (73.5)	
Declined/Unknown	0 (0)	5 (4.9)	9 (8.8)	
Smoking status, n (%)				
Current smoker	2 (3.9)	4 (3.9)	0 (0)	0.05
Former smoker	4 (7.8)	14 (13.6)	8 (7.8)	
Never smoker	42 (82.4)	43 (41.7)	38 (37.3)	
Unknown	3 (5.9)	42 (40.8)	56 (54.9)	
Rupture of membranes, n (%)				
Artificial	42 (82.4)	50 (48.5)	49 (48)	1
Spontaneous	9 (17.6)	52 (50.5)	53 (52)	
Delivery method, n (%)				
Cesarean	7 (13.7)	58 (56.3)	68 (66.7)	0.17
Vaginal	44 (86.3)	45 (43.7)	34 (33.3)	
Twin delivery, n (%)	2 (3.9)	33 (32)	30 (29.4)	0.80
Antenatal steroids, n (%)				
None	51 (100)	6 (5.8)	6 (5.9)	0.32
Complete	0 (0)	83 (80.6)	74 (72.5)	
Incomplete	0 (0)	14 (13.6)	22 (21.6)	
Preterm labor	0 (0)	75 (72.8)	64 (62.7)	0.16
Preterm premature rupture of membranes	0 (0)	51 (49.5)	48 (47.1)	0.83
Prolonged rupture of membranes	3 (5.9)	31 (30.1)	31 (30.4)	1
Pre-eclampsia/HELLP	1 (2)	17 (16.5)	27 (26.5)	0.14
Antenatal IUGR	0 (0)	2 (1.9)	16 (15.7)	0.001
Non-reassuring fetal heart tones	3 (5.9)	14 (13.6)	19 (18.6)	0.43
Chorioamnionitis	49 (96.1)	73 (70.9)	68 (66.7)	0.62
Diabetes, n (%)				
None	49 (96.1)	98 (95.1)	94 (92.2)	0.67
Gestational	2 (3.9)	4 (3.9)	6 (5.9)	
Insulin-dependent	0 (0)	1 (1)	2 (2)	
INFANT				
Gestational age, weeks, (mean ± SD)	39.5 ± 1.0	27.7 ± 1.2	26.1 ± 1.6	<0.001
Gestational age category, n (%)				
22–24 weeks	-	1 (1.0)	25 (24.5)	<0.001
25–26 weeks	-	25 (24.3)	39 (38.2)	
27–29 weeks	-	77 (74.8)	38 (37.3)	
Infant sex, n (%)				
Female	25 (49)	45 (43.7)	52 (51)	0.37
Male	26 (51)	58 (56.3)	50 (49)	
Birth weight (g), (mean ± SD)	3430 ± 452	1110 ± 201	854 ± 202	<0.001
Birth-weight-for-GA percentile (%)	52.8 ± 26.3	64.3 ± 21.2	56.3 ± 28.4	0.02
Small-for-gestational-age status, n (%)	2 (3.9)	1 (1)	9 (8.8)	0.02
APGAR score, median [min, max]				<0.001
1-min	8 [2, 9]	6 [0, 9]	4 [0, 8]	
5-min	9 [8, 9]	8 [1, 9]	7 [1, 9]	
BPD grade, n (%)				
Grade I	-	0 (0)	33 (32.4)	--
Grade II	-	0 (0)	26 (25.5)	
Grade III/IIIA	-	0 (0)	43 (42.2)	
Other neonatal complications, n (%)				
Necrotizing enterocolitis (any)	0 (0)	6 (5.8)	24 (23.5)	<0.001
Intraventricular hemorrhage (any)	0 (0)	24 (23.3)	42 (41.2)	<0.001
Retinopathy of prematurity (stage 2+)	0 (0)	9 (8.7)	32 (31.4)	<0.001
Pulmonary hypertension @ 36 weeks	0 (0)	8 (7.8)	28 (27.5)	<0.001

* *p*-value calculated comparing BPD versus non-BPD groups, using Chi-square or Fisher’s exact tests for categorical data, and parametric/non-parametric tests where appropriate.

## Data Availability

The raw adductomics data and the coded (deidentified) clinical data generated in this study are available from the corresponding authors upon request.

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
