# Peer review of "Cord Blood Adductomics Reveals Oxidative Stress Exposure Pathways of Bronchopulmonary Dysplasia"

_antioxidants, 2024, doi:10.3390/antiox13040494_

Round 1

Reviewer 1 Report

The idea of presenting adductomics applied with HSA-Cys34-related high-resolution technique to elucidate exposure pathways of preterm newborns is a promising approach in BPD research, which could be useful for the future identification, classification, and management of BPD and its subphenotypes

All figures are well described and self-explanatory; the statistical analysis is well developed and the results are clear and unambiguous. In this study, Lin E.T et al. applied adductomic to investigate protein adducts in cord blood and peripheral blood in extreme preterm (EPT) (n=205) newborns with or without Bronchopulmonary Dysplasia BPD. Similarly, the authors compared peripheral blood adducts in 1-week, 1-month, and 36-weeks postmenstrual age infants. HSA was isolated from plasma with or without BPD. HPLC-mass spectrometry was applied to identify, quantify, and compare the presence of a group of already known adducts (small thiols, direct oxidation products, reactive aldehydes) (n=51), but, more interestingly, of a number of not yet known adducts. The marker used for the detection was human serum albumin or the HSA-Cys34 protein, a more reliable biomarker than those commonly used for the detection of the state of oxidative stress, since the half-life of the HSA-Cys34 protein is approximately 4 weeks in plasma.

Results showed that the 51 known adducts were increased with BPD depending on the degree of prematurity and that in infants monitored after 1-week, 1month and after 38-weeks-post-menstruation ages, serial concentrations of several known adducts correlated directly with supplemental oxygen exposure.

 The introduction describes the background of the study focusing on the three main topics: BPD, ROS and the approach used, i.e. adductomics in the presence of HSA-Cys34. The bibliography reported is relevant.

The methods are adequately described and figure 1, which describes how the analyzed samples were divided and what all the investigative steps were, contributes to the clarity of the work carried out.

The results are presented with a logical succession, starting from the characterization of the samples (supported by adequate clinical and demographic investigation). The classification of the adducts (supplementary data) with the distinction between known and unknown is clear and is taken up again in subsequent comparisons between the samples of the various groups, both EPT and FT. All supported by a robust statistical analysis (linear regression applied to verify the differences between adducts between FT and BPD and non-BPD is adequate). Significance derived from Bonferroni p value is accepted.

The only objection that could be brought to the study could be the incorrect or partial evaluation of the gestational conditions that had led to the premature birth, such as preeclampsia, chorioamnionitis, diabetes, or other. Instead, the authors presented an evaluation of the adducts also based on these clinical notes, thus arriving at a complete analysis of the various factors that could influence the increase in known/unknown adducts.

 Lastly, the evaluation of the effect of the care given to the children (cumulative supplemental oxygen, CSO) was monitored over time and reported in 46 infants.

The discussion resumes the results obtained without being verbose or repetitive, from which it subsequently extends to cover the pre-established field of interest, i.e. the evidence of the need for new tools for the diagnosis of BPD and, above all, for the evaluation of correct administration of CSO in infants to prevent its side effects, BPD.

Finally, the need to apply new studies with this or other methodologies to study the problem is also underlined given the (relative) limitation of the numerical sample analyzed.

Author Response

Reviewer #1:

  1. “The only objection that could be brought to the study could be the incorrect or partial evaluation of the gestational conditions that had led to the premature birth, such as preeclampsia, chorioamnionitis, diabetes, or other…” Instead, the authors presented an evaluation of the adducts also based on these clinical notes, thus arriving at a complete analysis of the various factors that could influence the increase in known/unknown adducts.

We greatly appreciate this reviewer’s comments. In the original draft we included more comprehensive detail of the covariates of preterm birth (specifically, preeclampsia and chorioamnionitis). These risk factors were stringently assessed prospectively in our birth cohort using standardized study criteria based upon ACOG guidelines and our cohort’s placental examination criteria.  We have added these details back into the revised methods (lines 103-110) and also included a paragraph in the results (Section 3.2.2., lines 251-263) detailing our analysis of preeclampsia and chorioamnionitis. As shown in Table 1, diabetes was not a common complication in this cohort so we did not conducted subgroup analysis accordingly.

Reviewer 2 Report

The paper entitled “Cord Blood and Postnatal Adductomics Reveal Novel Exposure Pathways of Bronchopulmonary Dysplasia” by Erika T. Lin et al. describes the adductomics analysis of fetal and neonatal blood and correlates analytical results with patients' exposures to perinatal oxidative stress (OS).

The application of  the novel adductomics approach has the potential of novel discoveries in biomedicine and, as such, it should be welcomed.

Moreover, authors performed both untargeted and targeted adductomics, which adds value to the study from an analytical point of view.

However, for this study to provide interest in the biological, biomedical or antioxidants fields, it should address specific issues. Biochemical and biological conclusions appear very preliminary, instead, according to the analysis of data here presented.

In particular, the claim “novel exposure pathways” appear a little far fetched, based on the presented elaboration.

We think clarification of the following points could help improve the paper and strengthen its conclusions.

  1. the title is confusing at a first glance. It deals with bronchopulmonary dysplasia, a specific condition, certainly correlated to perinatal oxidative stress but not exclusively so. 

  2.  The abstract is confusing as well. BPD is not defined and put there in a snap. An introductory phrase about this could help the reader in approaching the content. The flow could be:   oxidative stress suffered during birth generates oxidative stress markers that are revealed by exposomics analysis (novel markers have been discovered??).

  3. The Introduction clarifies better the study design. However, known (and eventually novel) oxidative stress pathways explored remain undefined. This is probably why the claim about novel exposure pathways appears out of place from the beginning to the end.

  4. A comparison of the results obtainable and then obtained by targeted and untargeted complementary approaches could be useful, provided that both were performed.

  5. In methods:

  1. Since neonatal blood drawing is an invasive procedure, could dried blood spots be used for exposomics analysis?

  1. Why the ALVLIA… peptide was used? Please explain at least briefly

In results:

Sampling time points ..how were they correlated between groups. Were they the same? Or comparable? A time table or a summary of corresponding timepoints between groups could help get at a glance the presence of oxidative stress markers.

Figures are low resolution and printed parameters unreadable in Fig.4 - 7

In discussion, please try to provide an answer to issues such:

What markers are novel or specific to peculiar oxidative stress conditions examined, among those detected?

What degree of confidence was associated with each potential biomarker?

Could some of them be used in diagnostics laboratories? 

Or is the analysis aimed at discovering novel biological mechanisms? In that case, which are more relevant to the peculiar study? 

Could clinical interventions be guided by analyzed parameters?

Author Response

Reviewer #2:

We thank this reviewer for their careful review and thoughtful comments.  Below we provide point-by-point responses to the changes made in the revised manuscript. 

Major comments

  1. “…Biochemical and biological conclusions appear very preliminary, instead, according to the analysis of data here presented…In particular, the claim “novel exposure pathways” appear a little far fetched, based on the presented elaboration.”

Response:  We agree with this reviewer that “novel pathways” does not meet the expectation of the contents of this manuscript.  While the overall approach of identifying exposure biomarkers and pathways in study BPD is novel, the pathways identified are not necessarily novel as they have been described (or in the case of untargeted adducts the pathways are not yet understood). The concept of linking oxidant stress to BPD through adducts is novel, as this has not previously been done.  Thus, we have revised the title de-emphasizing “novel” and emphasizing “oxidant stress exposure pathways.”

We have also revised several areas on the manuscript to clarify that the pathways themselves are not novel.

Detailed comments

  1. “…the title is confusing at a first glance. It deals with bronchopulmonary dysplasia, a specific condition, certainly correlated to perinatal oxidative stress but not exclusively so…”

Response:  We have modified the title to read, “Cord Blood Adductomics Reveals Oxidant Stress Exposure Pathways of Bronchopulmonary Dysplasia.”  Given the scope of this special edition of the role of antioxidants in pregnant women's and children's health, we feel this revised title is more representative of the work described. Also, to keep the title within length, we have removed references to focus on cord blood, which better describes the main scope of this work.

2.  “The abstract is confusing as well. BPD is not defined and put there in a snap. An introductory phrase about this could help the reader in approaching the content…”

Response:  The abstract has been revised to set the stage for perinatal oxidant stress as a key mediator of BPD, followed by the rationale for studying adductomics:

“Fetal and neonatal exposures to perinatal oxidative stress (OS) are key mediators of bronchopulmonary dysplasia (BPD). To reliably characterize these exposures…”

3.  “The Introduction clarifies better the study design. However, known (and eventually novel) oxidative stress pathways explored remain undefined. This is probably why the claim about novel exposure pathways appears out of place from the beginning to the end.

Response:  We thank the reviewer for these recommendations.  We have removed reference to “novel exposure pathways” in the Introduction (Line 67) and replaced it with “oxidant stress-related exposure pathways” to address this reviewer’s concerns, consistent with the revised title. Please also see response to #1 regarding changes throughout the manuscript. To clarify, the primary objective of this study is to identify targeted and untargeted adducts associated with BPD. Upon further characterization (in future studies) these adducts could potentially be used to guide BPD management. This has been clarified in the Introduction in Lines 67-75 by removing reference to elucidating intrauterine exposure pathways.

4.  A comparison of the results obtainable and then obtained by targeted and untargeted complementary approaches could be useful, provided that both were performed.

Response:  To clarify the approach, we have added the below paragraph describing the preliminary studies performed that characterized targeted and untargeted adducts measured in this study (Lines 164-169):

“Untargeted adductomics were initially performed on (n= samples) across the gestational age spectrum (32 to 40 weeks). This served to identify all known adducts with corresponding structures as well as detect unknown adducts with new organic structures for analysis. Once all adducts were validated, targeted adductomics results could be compared to validated known and unknown adducts from untargeted adductomics.” 

In methods:

1.  Since neonatal blood drawing is an invasive procedure, could dried blood spots be used for exposomic analysis?

Response:  As mentioned in Lines 495-497 of the original manuscript, dried blood spots provide promise for the development of more practical approaches to tracking exposure biomarkers over time in critically ill neonates.  To clarify further, we emphasize that an important advantage is that DBS approaches are non-invasive.  Our group has extensive experience in measuring DBS samples from peripheral blood for adductomics.  These references are highlighted in the revised manuscript.[REFS 53-57]  An important limitation for our study is that plasma and DBS samples are processed very differently and the output levels cannot be analyzed interchangeably.  Since cord blood dried blood spots are not routinely available in most delivery centers (including ours), the best approach at the inception of this project was to track plasma from birth (cord blood) through the neonatal period in peripheral blood.  These limitations are now mentioned in Lines 513-515 of the revised manuscript.

2.  Why the ALVLIA… peptide was used? Please explain at least briefly…

Response:  T3 peptide (sequence: ALVLIAFAQYLQQCPFEDHVK), one of the main products generated by tryptic digestion of human serum albumin (HSA), contains Cys34.  As mentioned in Lines 57-58, Cys34 can serve as a nucleophilic hotspot for protein adducts.  We now briefly outlined the rationale for utilizing the T3 peptide in Lines 144-145:

“Here, it is important to note that the T3 peptide contains Cys34, which can promote the formation of protein adducts.”

In results:

1.  Sampling time points...how were they correlated between groups. Were they the same? Or comparable? A time table or a summary of corresponding timepoints between groups could help get at a glance the presence of oxidative stress markers.

Response:  The peripheral blood samples were collected at 1-week (median [IQR]=7 days [7, 7]), 1-month (30 days [30, 30]) and 36-weeks postmenstrual age (36 weeks [36, 36]). The median timepoints were the same for BPD and no-BPD groups.  In the supplemental Table S2, we now indicate the adduct levels according to BPD and no-BPD status for each timepoint in order to demonstrate how the presence of the markers varies across timepoints and between groups.  The median and ranges for BPD groups are highlighted by the red boxes so that trends over the 4 collection timepoints (cord blood at birth, 1-week, 1-month and 36 weeks) and compared to no-BPD groups at each timepoint are better visualized. 

2.  Figures are low resolution and printed parameters unreadable in Fig.4 – 7

Response:  Figures 4-7 have been revised to provide improved resolution, including the printed parameters.

In discussion:

  1. What markers are novel or specific to peculiar oxidative stress conditions examined, among those detected?

Response:  The types of oxidative stress conditions are shown in Figure 3.  The markers associated with these specific conditions are listed according to oxidant stress pathway in Supplemental Table 1, according to the following classifications:  1) Small Thiols are Cys34 disulfides formed with small thiol molecules related to antioxidant capacity; 2) Direct oxidation products are the addition of 1-3 oxygen molecules formed by reactions with Cys34 and reactive oxygen species. 3) Reactive aldehydes are reaction products formed through lipid peroxidation…

The top 6 conditions are indicated in Figures 4A and 5A, as I-VI on the left.  The adduct names (A034, A039…etc) are now more readable on the right in the higher resolution figures, and links each adduct to its oxidant stress classification.

2.  What degree of confidence was associated with each potential biomarker?

Response:  The median and 1st and 3rd quartile ranges are reported for each biomarker in Supplemental Table S2 for each of the conditions (Full-Term, No BPD, BPD, 1 week, 1-month, 36-weeks).  As this study was not aimed at quantifying exact biomarker levels as predictors of these conditions, but rather comparing fold-change, there were no confidence intervals calculated or reported.

3.  Could some of them be used in diagnostics laboratories? 

Response:  While this was not the scope of this project, some of the HSA-Cys34 biomarkers can potentially be used in diagnostics laboratories in the near future, as the technology for adductomics is a rapidly advancing.  As mentioned in the above response, Cys34 sulfinic acid (A012) can be formed through endogenous reaction with reactive oxygen species.  By training the medical staffs on how to pretreat blood samples (e.g., plasma or DBS), some of the adducts, including Cys34 sulfinic acid (A012), can be used for measuring oxidative stress status.  We have now added this in Lines 526-530.

“Additionally, HSA-Cys34 biomarkers can be utilized in diagnostics laboratories. For instance, Cys34 sulfinic acid (A012) can indicate the status of reactive oxygen species.[23] By training medical staffs on how to pretreat blood samples (e.g., plasma or DBS), certain adducts, including Cys34 sulfinic acid (A012), can be used to measure oxidative stress status.”

4.  Or is the analysis aimed at discovering novel biological mechanisms? In that case, which are more relevant to the peculiar study? 

Response:  We are pleased that this reviewer acknowledges that the primary goal of this study was to elucidate novel biological mechanisms of BPD.  While it is known that oxidant stress plays a role in the pathogenesis of lung injury and BPD, there are no reliable biomarkers to track the types of oxidant stress (e.g. lipid peroxidation, oxidative DNA damage), which of these processes are present at birth, and which processes persist over time with hyperoxia and other exposures. 

This study was focused more on investigating biological mechanisms rather than developing diagnostic tests.  However, the technology has advanced over time since the inception of this study, such that diagnostic utility may be considered in the near future.

5.  Could clinical interventions be guided by analyzed parameters?

Response:  Given the complex nature of caring for extremely preterm infants, clinical interventions cannot practically be guided by adductomics at this time.  In fact, there are very few blood biomarkers that are used in the NICU to guide manage, so the concept of this approach is novel.  The next steps in our investigations will be to specific adducts that could assess oxidant stress levels and provide interventions to reduce these levels. 

Round 2

Reviewer 2 Report

All raised points have been properly addressed or at least mentioned as study limitations 

figures and relative legends have been improved